# WITH GREAT BACKBONES COMES GREAT ADVERSARIAL TRANSFERABILITY

## ABSTRACT

Advancements in self-supervised learning (SSL) for machine vision have enhanced representation robustness and model performance, leading to the emergence of publicly shared pre-trained backbones, such as *ResNet* and *ViT* models tuned with SSL methods like *SimCLR*. Due to the computational and data demands of pre-training, the utilisation of such backbones becomes a strenuous necessity. However, employing backbones may imply adhering to the existing vulnerabilities towards adversarial attacks. Prior research on adversarial robustness typically examines attacks with either full (*white-box*) or no direct access (*black-box*) to the target model, but the adversarial robustness of models tuned on known pre-trained backbones remains largely unexplored. Furthermore, it is unclear which tuning configuration is critical for mitigating exploitation risks. In this work, we systematically study the adversarial robustness of models that use such backbones, evaluating $20,000$ combinations of tuning configurations, including fine-tuning techniques, backbone families, datasets, and attack types. To uncover and exploit vulnerabilities, we propose to use proxy models to transfer adversarial attacks, fine-tuning them with various configurations to simulate different levels of knowledge about the target. Our findings show that proxy-based attacks can outperform strong query-based *black-box* methods with sizeable budgets approaching the effectiveness of *white-box* methods. Critically, we construct a naive "backbone attack", leveraging only the shared backbone, and show that even it can achieve efficacy consistently surpassing *black-box* and closing in towards *white-box* attacks, thus exposing critical risks in model-sharing practices. Finally, our ablations reveal how tuning configuration knowledge impacts attack transferability.

## 1 INTRODUCTION

Machine vision models pre-trained with massive amounts of data, which utilise self-supervised tuning techniques (Newell & Deng, 2020) are shown to be robust and highly performing (Goyal et al., 2021a; Goldblum et al., 2024) feature-extracting backbones (Elharrouss et al., 2022; Han et al., 2022), which are further used in a variety of tasks, from classification (Atito et al., 2021; Chen et al., 2020b) to semantic segmentation (Ziegler & Asano, 2022). However, creating such backbones incurs substantial data annotation (Jing & Tian, 2020) and computational costs (Han et al., 2022), consequently rendering the use of such publicly available pre-trained backbones the most common and efficient solution for researchers and engineers alike. Prior research has focused on analysing safety and adversarial robustness in different settings w.r.t. knowledge of the target model weights, fine-tuning data, fine-tuning techniques and other tuning configurations – complete knowledge, i.e. *white-box* (Porkodi et al., 2018) vs. no knowledge, i.e. *black-box* (Bhambri et al., 2019).

Although in practice, an attacker can access partial knowledge (Lord et al., 2022; Zhu et al., 2022; Carlini et al., 2022) of how the targeted model was produced, i.e. original backbone weights, tuning recipe, etc., the adversarial robustness of models tuned on a downstream task from a given pre-trained backbone remains largely underexplored. We refer to settings with partial knowledge of the target model tuning configuration as *grey-box* (S. et al., 2018). These types of configurations are important both for research and production settings because with an increased usage (Goldblum et al., 2023) of publicly available pre-trained backbones for downstream applications, we are still incapable of assessing the potential exploitation susceptibility and inherent risks within models tuned on top of them and subsequently enhance future pre-trained backbone sharing practices.

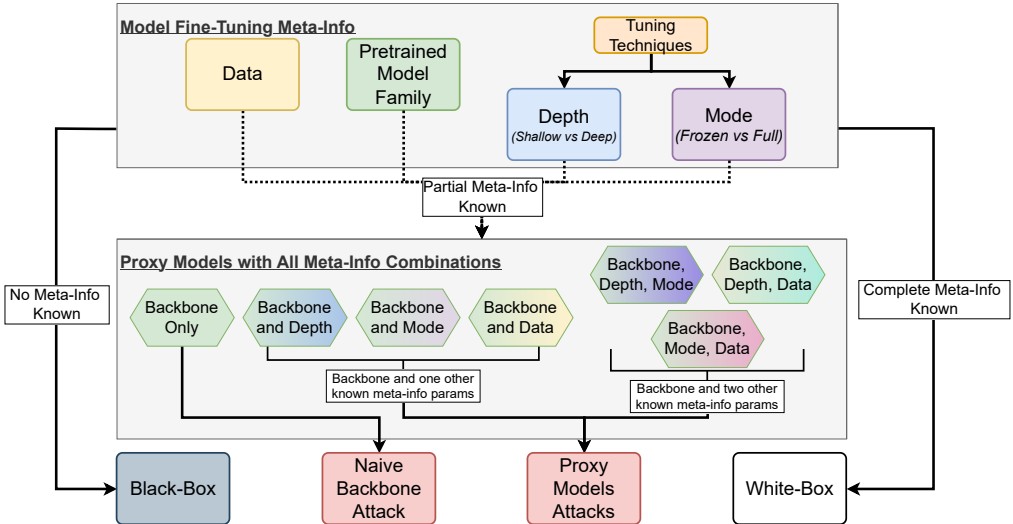

Figure 1: The figure depicts all of the settings used to evaluate adversarial vulnerabilities given different information of the target model construction. From left to right, we simulate exhaustive varying combinations of tuning configurations available from the target model during adversarial attack construction. All of the created proxy models are used separately to assess adversarial transferability.

To address this gap, in our work, we systematically explore the safety from adversarial attacks in models tuned on downstream classification tasks from known publicly available backbones pre-trained with self-supervised objectives. We further explicitly measure the effect of the target model construction configuration by simulating different levels of its availability during the adversarial attack. For this purpose, we initially train 352 diverse models from 21 families of commonly used pre-trained backbones using 4 different fine-tuning techniques and 4 datasets. We fix each of these networks as potential target models and transfer adversarial attacks using all other models produced from the same backbones as proxy surrogates (Qin et al., 2023; Lord et al., 2022) for the construction of adversarial attacks. Each surrogate model simulates varying levels of knowledge availability w.r.t. target model construction configuration on top of the available backbone during adversarial attack construction. This constitutes approximately 20, 000 adversarial transferability comparisons between target and proxy pairs across all model families and configuration variations. By assessing the adversarial transferability of attacks from these surrogate models, we are able to explicitly measure the impact of the availability of each combination of tuning configurations on the final target model during adversarial sample generation, as depicted in Figure 1.

We further explore a naive exploitation method referred to as *backbone attack* that only utilises the pre-trained feature extractor for adversarial sample construction, in this setting. The attack uses projected gradient descent over the representation space to disentangle the features of similar examples. Our results show that both proxy models and even simple *backbone attacks* are capable of surpassing strong query-based *black-box* methods and achieving comparable efficacy to *white-box* performance. The findings indicate that *backbone attacks*, where the attacker lacks knowledge of tuning configuration about the target model, are generally more effective than attempts to generate adversarial samples with limited knowledge. This highlights the vulnerability of models built on publicly available backbones.

Our ablations show that *having access to the weights of the pre-trained backbone is functionally equivalent to possessing all other tuning configurations about the target model when performing adversarial attacks*. We compare these two scenarios and show that both lead to similar vulnerabilities, highlighting the interchangeable nature of these knowledge types in attack effectiveness. Our results emphasise the risks in sharing and deploying pre-trained backbones, particularly concerning the disclosure of configurations. Our experimental framework can be seen in Figure 1.

In summary, our contributions are as follows: **(i)** we formalize and systematically study **grey-box** adversarial attacks, which reflects realistic scenarios where attackers have partial knowledge of target model tuning configuration, such as access to pre-trained backbone weights and/or fine-tuning configuration; **(ii)** we simulate over $20,000$ comparisons of adversarial transferability, evaluating the impact of varying levels of tuning configuration availability about target models during the construction of attacks; **(iii)** we explore a naive attack method, *backbone attacks*, which leverages the pre-trained backbone's representation space for adversarial sample generation, demonstrating that even such a simple approach can achieve stronger performance compared to query-based black-box methods and often approaching white-box attack effectiveness; **(iv)** we show that access to pre-trained backbone weights alone enables adversarial attacks as effectively as access to the full tuning configuration about the target model, emphasizing the inherent vulnerabilities in publicly available pre-trained backbones.

## 2 Related Work

**Self Supervised Learning**   With the emergence of massive unannotated datasets for machine vision such as YFCC100M (Thomee et al., 2016), ImageNet (Deng et al., 2009), CIFAR (Krizhevsky, 2009) and others, Self Supervised Learning (SSL) techniques (Jing & Tian, 2021) have become increasingly more popular for pre-training vision models (Newell & Deng, 2020). This prompted the creation of various families of SSL objectives, such as colorization prediction (Zhang et al., 2016), jigsaw puzzle solving (Noroozi & Favaro, 2016) with further invariance constraints (Misra & van der Maaten, 2020, PIRL), non-parametric instance discrimination (Wu et al., 2018, NPID, NPID++), unsupervised clustering (Caron et al., 2018), rotation prediction (Gidaris et al., 2018, RotNet), sample clustering with cluster assignment constraints (Caron et al., 2020, SwAV), contrastive representation entanglement (Chen et al., 2020a, SimCLR), self-distillation without labels (Caron et al., 2021, DINO) and others (Jing & Tian, 2021). Numerous architectures, like AlexNet (Krizhevsky et al., 2012), variants of ResNet (He et al., 2016) and visual transformers (Dosovitskiy et al., 2021; Touvron et al., 2021; Ali et al., 2021) were trained using these SSL methods and shared for public use, thus forming the set of widely used pre-trained backbones. We obtain all of these models trained with different self-supervised objectives from their original designated studies summarised in VISSL (Goyal et al., 2021b). An exhaustive list of models is shown in Table 3.

**Adversarial Attacks**   The availability of pre-trained backbones allows for testing them for vulnerabilities towards adversarial attacks, which are learnable imperceptible perturbations generated to mislead models into making incorrect predictions (Szegedy et al., 2014; Goodfellow et al., 2015). Several attack strategies have been studied, including single-step fast gradient descent (Goodfellow et al., 2014; Kurakin et al., 2017, FGSM), and computationally more expensive optimization-based attacks, such as projected gradient descent based attacks (Madry et al., 2018, PGD), CW (Carlini & Wagner, 2017), JSMA (Papernot et al., 2017), and others (Dong et al., 2018; Moosavi-Dezfooli et al., 2016; Madry et al., 2018; Ma et al., 2023). All of these attacks assume complete access to the target model, which is known as the *white-box* (Papernot et al., 2017) setting. These attacks can be *targeted* to confuse the model to infer a specific wrong class or *untargeted*, aiming to make them infer any incorrect label. However, an opposite setting with no information, referred to as *black-box* (Papernot et al., 2017), has also been explored as a more common setting during adversarial attack construction. These methods involve attempts at gradient estimation (Chen et al., 2017; Ilyas et al., 2018; Bhagoji et al., 2018), adversarial transferability (Papernot et al., 2017; Chen et al., 2020c), local search (Narodytska & Kasiviswanathan, 2016; Brendel et al., 2018; Li et al., 2019; Moon et al., 2019), combinatorial perturbations (Moon et al., 2019) and others (Bhambri et al., 2019). However, a great portion of these methods also require massive sample query budgets ranging from $\left[10^3, 10^5\right]$ queries, or computational resources for creating each adversarial sample (Bhambri et al., 2019). Compared to these, we introduce a novel setup with the knowledge of the pre-trained backbone and varying levels of partially known target model tuning configuration during adversarial attack construction, which we refer to as *grey-box*. This setup reflects common scenarios where attackers have partial knowledge of the target model tuning configuration, allowing them to systematically assess the effect of this knowledge on adversarial transferability and show the risks in the current model-sharing practices. We show that even simple naive attacks are more capable of exploiting models without the need for a sizable query budget compared to black-box attacks.

**Adversarial Transferability**   Our work is also aligned with adversarial transferability, where adversarial examples generated for one model can mislead other models, even without access to the target model weights or training data. This property poses significant security concerns, as it allows for effective black-box attacks on systems with no direct access (Papernot et al., 2017; Ilyas et al., 2018). Efforts can be divided into *generation-based* and *optimisation* methods. Generative methods have emerged as an alternative to iterative attacks, where adversarial generators are trained to produce transferable perturbations. For instance, Poursaeed et al. (2018) employs autoencoders trained on white-box models to generate adversarial examples. Most attacks aiming at adversarial transferability strongly depend on the availability of data from the target domain (Carlini & Wagner, 2017; Papernot et al., 2017), although attempts at improving the transferability of baseline adversarial samples have also been explored (Li et al., 2020; Zhang et al., 2022; Li et al., 2023; 2024; Naseer et al., 2020a). However, although current adversarial transferability methods claim to produce massive vulnerabilities in machine vision models, Katzir & Elovici (2021) examines the practical implications of adversarial transferability, which are frequently overstated. That study demonstrates that it is nearly impossible to reliably predict whether a specific adversarial example will transfer to an unseen target model in a black-box setting. This perspective shows the importance of systematically evaluating transferability in realistic settings, including scenarios where attackers are sensitive to the cost of failed attempts. In our study, we offer a novel systematic approach to explicitly assess the adversarial transferability with varying levels of configuration knowledge.

## 3 METHODOLOGY

**Preliminaries**   For consistency, we employ the following notation. We denote each dataset as $\mathcal{D} = \{\mathcal{X}, \mathcal{Y}\}$; where $\mathcal{X} = \{x_1, \ldots, x_{|\mathcal{D}|}\}$ is a set of images, with $x_i \in \mathcal{R}^{H \times W \times C}$; where $H, W$ and $C$ are the height, width and the channels of the image accordingly and $\mathcal{Y} = \{y_1 \ldots y_n\}$ is used as the set of ground truth labels. We denote the training, validation and testing splits per task as $\mathcal{D} = \{\mathcal{D}_{train}, \mathcal{D}_{val}, \mathcal{D}_{test}\}$. A *model* is defined as a tuple $\mathcal{M} = \mathcal{M}(\mathcal{D}, \mathcal{W}, \mathcal{B}, \mathcal{F})$, where $\mathcal{D}$ contains the dataset used for training, $\mathcal{W}$ are the weights of the trained model and $\mathcal{B}$ is the pre-trained back-bone $\mathcal{B}(\mathcal{W}_{\mathcal{B}})$ with available weights $\mathcal{W}_{\mathcal{B}}$. The notation $\mathcal{F}(\mathcal{T}, \mathcal{Z})$, where $\mathcal{T}$ encodes the *mode* of tuning (e.g., full fine-tuning, partial fine-tuning, etc.) and $\mathcal{Z}$ the *depth* of tuning of the final classifier on top of the backbone.

**Tuning configuration variations**   We define the variations of the available configuration about the target model $\mathcal{M}$ during an adversarial attack as a *unit of release* $\mathcal{R} = \mathcal{R}(\mathcal{M}(\mathcal{D}, \mathcal{W}, \mathcal{B}(\mathcal{W}_{\mathcal{B}}), \mathcal{F}(\mathcal{T}, \mathcal{Z})))$. For example, if the target fine-tuning mode $\mathcal{Z}^{\text{target}}$ and dataset $\mathcal{D}^{\text{target}}$ are not known, the unit of release will be $\mathcal{R} = \mathcal{R}(\mathcal{M}(*, \mathcal{W}, \mathcal{B}(\mathcal{W}_{\mathcal{B}}), \mathcal{F}(\mathcal{T}, *)))$. Note that the *black-box* setting will correspond to the unit of release $\mathcal{R}(\mathcal{M}(*, *, *, *, *))$ and the *white-box* setting to $\mathcal{R}(\mathcal{M}(\mathcal{D}, \mathcal{W}, \mathcal{B}(\mathcal{W}_{\mathcal{B}}), \mathcal{F}(\mathcal{T}, \mathcal{Z})))$, all the variations between these are considered *grey-box*. When discussing any experiments within the *grey-box* setup, we assume the minimal unit of release contains knowledge about at least the pre-trained backbone i.e. $\mathcal{R}(\mathcal{M}(*, *, \mathcal{B}(\mathcal{W}_{\mathcal{B}}), *))$.

**Adversarial Attacks with Proxy Models**   To test the adversarial robustness of the models trained from the same pre-trained backbone, we create a set of proxy models $\mathcal{M}^{\text{proxy}} = \{\mathcal{M}_1^{\text{proxy}} \ldots \mathcal{M}_v^{\text{proxy}}\}$ given the pre-trained backbone $\mathcal{B}$, where $v$ is the number of all possible units of release between *black-box* and *white-box* settings that include the backbone. For each proxy model $\mathcal{M}_i^{\text{proxy}}$ with its designated configuration unit of release $\mathcal{R}_i$, we use an adversarial attack $\mathcal{A}$ to generate adversarial noise and further transfer it to the target model $\mathcal{M}^{\text{target}}$. This means that given an example image $x$ with a label $y$, target and proxy models $\mathcal{M}^{\text{target}}, \mathcal{M}^{\text{proxy}}$ we want to produce a sample $x'$ that would fool the target model, such that $\arg\max \mathcal{M}^{\text{target}}(x') \neq y$. If we are using a targeted attack, we want $\mathcal{M}^{\text{target}}(x') = t$ where $t$ is the targeted class different from the ground truth $t \neq c_{gt}$. After creating the adversarial attack for each sample in $\mathcal{D}_{\text{test}}^{\text{proxy}}$ and $\mathcal{D}_{\text{test}}^{\text{target}}$, we evaluate the success rate of the attack and the success rate of the transferability to the target model. To measure the success and robustness of the adversarial attack and its transferability, we define the following metrics:

**Attack Success Rate (ASR).** The proportion of adversarial examples that fool the proxy model $\mathcal{M}_i^{\text{proxy}}$:

$$\text{ASR}_i = \frac{1}{|\mathcal{D}_{\text{test}}^{\text{proxy}}|} \sum_{x \in \mathcal{D}_{\text{test}}^{\text{proxy}}} \mathbb{I}\left[\mathcal{M}_i^{\text{proxy}}(x') \neq y\right], \tag{1}$$

---

**Algorithm 1** Backbone Attack

---

**Input:** Model backbone $\mathcal{B}$, clean image $x_0$, perturbation bound $\epsilon$, step size $\alpha$, number of steps $T$, distance function $\mathcal{L}_{\text{cosine}}$, random start flag
**Output:** Adversarial image $x_{\text{adv}}$
**Initialization:**
  $x_{\text{adv}} \leftarrow x_0$
**if** *random start* **then**
    $x_{\text{adv}} \leftarrow x_{\text{adv}} + \text{Uniform}(-\epsilon, \epsilon)$
     $x_{\text{adv}} \leftarrow \text{Clip}(x_{\text{adv}}, 0, 1)$
**end**
**Fixed Original Image Representation:**
  $z_0 \leftarrow StopGrad(\mathcal{B}(x_0))$

**for** $t = 1$ **to** $T$ **do**
    **Forward Pass:**
      $z_{\text{adv}} \leftarrow \mathcal{B}(x_{\text{adv}})$ `// Adversarial image representation`
    **Compute Loss and Gradient:**
      $\mathcal{L} \leftarrow 1 - \cos(z_{\text{adv}}, z_0)$ `// Distance loss`
    $g \leftarrow \nabla_{x_{\text{adv}}} \mathcal{L}$ `// Gradient w.r.t` $x_{\text{adv}}$
    **Update Adversarial Image:**
      $x_{\text{adv}} \leftarrow x_{\text{adv}} + \alpha \cdot \text{sign}(g)$ `// PGD step`
    **Projection:**
      $\delta \leftarrow \text{Clip}(x_{\text{adv}} - x_0, -\epsilon, \epsilon)$ `// Project perturbation into` $\ell_\infty$`-ball with`
      `perturbation budget` $\epsilon$
    $x_{\text{adv}} \leftarrow \text{Clip}(x_0 + \delta, 0, 1)$ `// pixel range`
**end**
**return** $x_{\text{adv}}$

---

where $\mathbb{I}[\cdot]$ is the indicator function.

**Transfer Success Rate (TSR).** The proportion of adversarial examples generated by $\mathcal{M}_i^{\text{proxy}}$ that also fool the target model $\mathcal{M}^{\text{target}}$:

$$\text{TSR}_i = \frac{1}{|\mathcal{D}_{\text{test}}^{\text{target}}|} \sum_{x \in \mathcal{D}_{\text{test}}^{\text{target}}} \mathbb{I}\left[\mathcal{M}^{\text{target}}(x') \neq y\right]. \tag{2}$$

This setup allows us to explicitly quantify how the availability of diverse configuration combinations explicitly impacts the adversarial transferability of the given model, thus highlighting the risks in the model-sharing practices. A visual depiction of this can be seen in Figure 1.

## 3.1 BACKBONE ATTACK

To test the vulnerabilities associated with publicly available pre-trained feature extractors, we construct a *backbone attack*, which only utilises the known backbone $\mathcal{B}$ of the model $\mathcal{M}^{\text{target}}$. The aim, similar to the prior paragraph, is to create an adversarial attack from $\mathcal{B}$ to transfer to the target model $\mathcal{M}^{\text{target}}$. To do this, we use a Projected Gradient Descent-based method (Madry et al., 2018, PGD), where the attack iteratively perturbs the input images in order to maximize the distance between the feature representations of the clean input and the adversarial input, as derived from the backbone $\mathcal{B}$. More formally, let $x$ and $\tilde{x}$ represent the clean input and adversarial input, respectively. The attack iteratively refines $\tilde{x}$ such that:

$$\tilde{x}_{t+1} = \text{Proj}_{\mathcal{S}}\left(\tilde{x}_t + \alpha \cdot \text{sign}\left(\nabla_{\tilde{x}_t} \mathcal{L}_{\mathcal{B}}(x, \tilde{x}_t)\right)\right), \tag{3}$$

where $\mathcal{L}_{\mathcal{B}}$ is the loss function defined to measure the distance between the feature representations of the clean and adversarial inputs. The backbone representations $f_{\mathcal{B}}$ are extracted as $f_{\mathcal{B}}(x) = \mathcal{B}(x)$, and the differentiable loss can be formulated as:

$$\mathcal{L}_{\mathcal{B}}(x, \tilde{x}) = 1 - \cos\left(f_{\mathcal{B}}(x), f_{\mathcal{B}}(\tilde{x})\right), \tag{4}$$

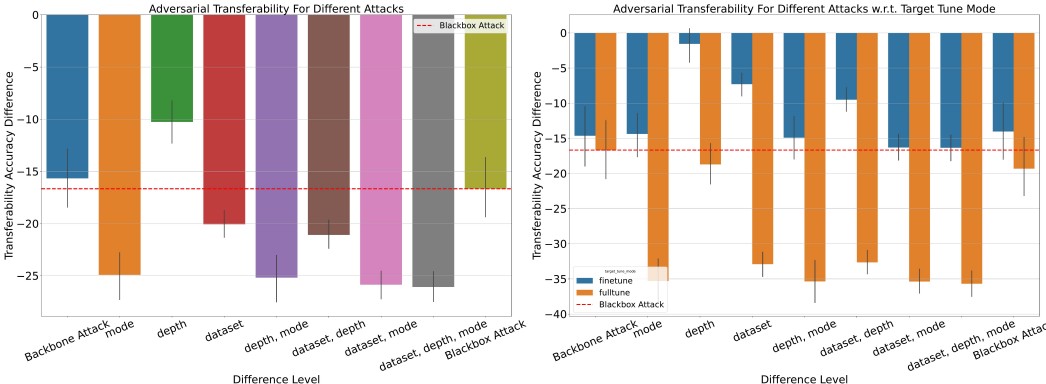

Figure 2: The impact of the **unavailability**, i.e. difference from the target model *white-box* performance, of all tuning configurations on adversarial transferability during proxy attack construction and the backbone attack. The results show the average difference from the *white-box* in transferability using PGD with a higher budget (left) and the segmentation w.r.t. in the target training mode (right).

where $\cos(\cdot, \cdot)$ represents the cosine similarity between the two feature vectors. To prevent gradient computation from propagating to the clean representation $f_{\mathcal{B}}(x)$, we utilize a stop-gradient operation $\tilde{f}_{\mathcal{B}}(x) = \text{SG}(f_{\mathcal{B}}(x))$. The adversarial input $\tilde{x}$ is initialized with a random perturbation within the $\ell_{\infty}$ ball of radius $\epsilon$, and the updates are iteratively projected back onto this ball using the $\text{Proj}_{\mathcal{S}}$ operator:

$$\text{Proj}_{\mathcal{S}}(\tilde{x}) = \text{clip}\,(x + \delta, 0, 1)\,, \tag{5}$$
$$\text{where} \quad \delta = \text{clip}\,(\tilde{x} - x, -\epsilon, \epsilon)\,.$$

The pseudo-code of the complete process can be seen in Algorithm 1. In summary, the backbone attack focuses solely on the backbone $\mathcal{B}$, without requiring any knowledge of the full target model $\mathcal{M}^{\text{target}}$, thereby revealing vulnerabilities inherent to publicly available feature extractors. A form of this algorithm has been utilised as a naive self-supervised perturbation generation component in adversarial defence training (Naseer et al., 2020b, NPR), however, it has not been explored individually. We only use this attack to showcase that even naive backbone exploitation methods can have significant adversarial transferability.

## 4 EXPERIMENTAL SETUP

**Image classification datasets**
Through our study, we use 4 datasets covering both classical and domain-specific classification benchmarks, such as CIFAR-10 and CIFAR-100 (Beyer et al., 2020), Oxford-IIIT Pets (Parkhi et al., 2012) and Oxford Flowers-102 (Nilsback & Zisserman, 2008). We train the proxy and target model variations on each one of the datasets using the recipe and hyperparameters by (Kolesnikov et al., 2020), reproducing the state-of-the-art model performance results (Dosovitskiy et al., 2020; Yu et al., 2022; Bruno et al., 2022; Foret et al., 2020).

| Metadata type | Original Entropy | | Adversarial Entropy | |
|---|---|---|---|---|
| | **F-Statistic** | **P-Value** | **F-Statistic** | **P-Value** |
| *Target Tune Mode* | 0.00 | 0.96 | 1238.7 | 0.0 |
| *Proxy Tune Mode* | 0.02 | 0.88 | 0.5 | 0.4 |
| *Target Dataset* | 2812.25 | 0.00 | 1184.1 | 0.0 |
| *Proxy Dataset* | 8.31 | 0.00 | 5.0 | 0.0 |
| *Target Tune Depth* | 5.64 | 0.01 | 0.36 | 0 |
| *Proxy Tune Depth* | 0.08 | 0.77 | 0.00 | 0 |

Table 1: Variance analysis of entropy values across categorical variables. The table shows F-statistics and p-values for both original and adversarial entropy means. Significant p-values ($p < 0.05$) show notable variations in entropy across tuning configurations.

**Model variations** We use 21 different models tuned from 5 architectures, 9 self-supervised objectives and 3 pre-training datasets. A detailed overview of these can be seen in Table 3 in Section A.1.

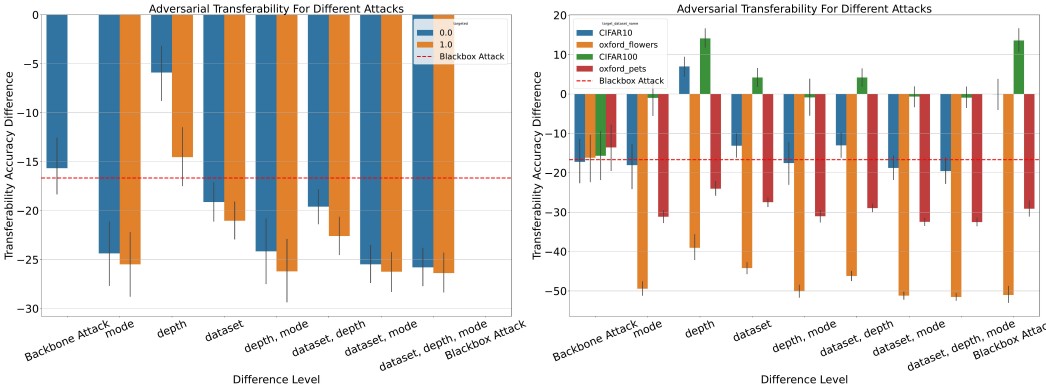

Figure 3: The impact of the **unavailability**, i.e. difference from the target model *white-box* performance, of all tuning configurations on adversarial transferability during proxy attack construction and the backbone attack. The results show the average transferability for PGD with a higher budget for targeted vs untargeted attacks (left) and the segmentation w.r.t. the target training dataset (right).

**Model Fine-tuning Variations**    For training the proxy and target models, we employ two *modes* of training $\mathcal{T}$, with full-tuning of the weights and with fine-tuning only the last added classification layers on top of the pre-trained backbone. We also define the depth of tuning $\mathcal{Z}$ as the number of classification layers added on top of the pre-trained backbone. We use $\{1, 3\}$ final layers, corresponding to *shallow* and *deep* tuning settings.

**Adversarial Attacks**    To assess the success rate of *white-box* adversarial attacks and the adversarial transferability from the proxy models, we employ FGSM (Goodfellow et al., 2015) and PGD (Madry et al., 2018). We use standard attack hyper-parameters introduced in parallel adversarial transferability studies (Waseda et al., 2023; Naseer et al., 2022). For a fair comparison, we also use the same values for our *backbone-attack*. We also impose a standard perturbation budget $\epsilon \leq \frac{8}{255}$ in line with prior studies (Naseer et al., 2022) outline in Algorithm 1. To show that our results are consistent even with a higher computational budget, we report the results of PGD with $4$ times more iterations per sample for *white-box*, proxy and *backbone* attack experiments. For *black-box* experiments, we use the Square attack (Andriushchenko et al., 2020), which is a query-efficient method that uses a random search through adversarial sample construction. To standardise the query budget for all architectures and simulate real-world constraints, we allow 10 queries of the target model per sample. The information about the used computational resource can be found in Section A.2.

## 5    RESULTS

### 5.1    WHAT CONFIGURATION MATTERS

To quantify the impact of each possible configuration availability along with the backbone knowledge during adversarial attack construction, we compute the difference between the adversarial attack success rate (ASR) for the target model and the transferability success rate (TSR) from a proxy model, trained from the same backbone, with partial information. We report the results obtained with the PGD attack trained with higher iteration steps per sample as that is more representative for measuring the adversarial attack success in *white-box* and *grey-box* settings.

**Which configuration is important?**    Our results in Figure 2 show that the most significant performance decay compared to a *white-box* attack performance occurs when the attacker is unaware of the *mode* of the training of the target model, i.e. if it is trained with complete parameters or only tunes the last classification layers. The second most impactful knowledge for attack construction is the availability of the target tuning *dataset*. The *depth* of the tuning is the least important knowledge for obtaining a transferable attack. We further show in the right part of Figure 2 that models that fine-tune the last classification layers can be trivially exploited with transferable attacks, achieving results significantly better than strong black-box exploitation and closing white-box attack performance. It

is, however, apparent that training all of the model weights substantially decreases the efficiency of proxy attacks, with almost no correlation towards configuration availability. We further show that our results remain consistent w.r.t. the choice of the dataset, and regardless if the adversarial attack is targeted or untargeted as seen in Figure 3. It is interesting to note that for datasets with more domain-specific content, such as Oxford-IIIT Pets and Oxford Flowers-102, the effectiveness of the proxy attack dwindles, although these datasets are much less diverse compared to CIFAR-100.

**Tuning configuration impacts the quality of adversarial attacks** We also want to measure the effectiveness of the adversarial attack and the impact of the tuning configuration on it by quantifying how the generated adversarial sample has shifted the decision-making of the model. To do this, we compute the entropy of the final softmax layer for each original sample and its adversarial counterpart, and perform a complete ANOVA variance analysis (St et al., 1989) of the entropy distribution. This analysis, presented in Table 1, tests whether the means of entropies from the original and adversarial images differ significantly across the groups of available tuning configurations. A perfect attack would produce a sample that does not majorly impact the entropy of the model. The analysis reveals that the target dataset, and tuning mode significantly influence entropy, particularly in adversarial scenarios. This finding sug-

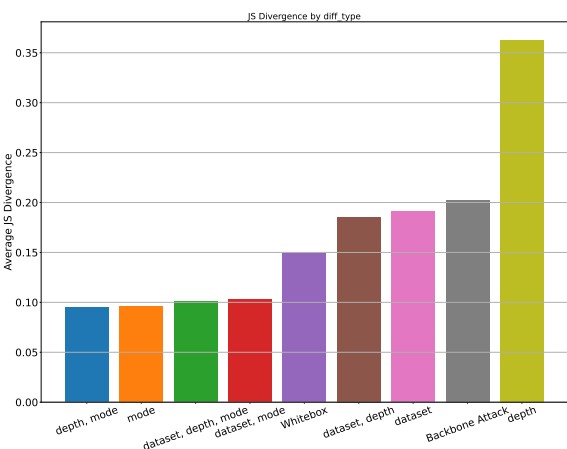

Figure 4: Impact of the **unavailability** of each tuning configuration on model decision-making. Higher JS divergence implies a bigger change in final classification.

gests that while this configuration aids in crafting effective adversarial samples, it also plays a critical role in amplifying entropy shifts, thereby making these adversarial samples more detectable.

To quantify the impact of the availability of tuning configuration during the construction of attacks on the decision-making of the model, we also compute the Jensen-Shannon Divergence (Menéndez et al., 1997) between the output softmax distributions of the model produced for original samples and their adversarial counterparts, seen in Figure 4. High JS divergence suggests a strong attack, as the adversarial example causes a significant shift in the model's predicted probabilities, with minimal changes to the input sample under an imposed perturbation budget $\epsilon$. Our results show that not knowing the *mode* of the target model training causes the most degradation in constructing successful adversarial samples with proxy attacks. The second most important fact is the choice of the target *dataset*, while the *depth* of the final classification layers does not seem to be impactful for creating adversarial samples. Figure 4 reveals a critical insight: proxy attacks, even when constructed without knowledge of the target model's *dataset* or *depth*, can generate adversarial samples that induce more pronounced distribution shifts than *white-box* attacks. In other words, attackers do need to have access to the training dataset or model classification depth to craft adversarial samples capable of significantly disrupting the target model's decision-making process.

## 5.2 BACKBONE-ATTACKS

To test the extent of the vulnerabilities that the knowledge of the pre-trained backbone can cause, we evaluate a naive exploitation method, *backbone attack*, which only uses the pre-trained feature extractor for adversarial sample construction. Our results in Figure 2 and Figure 3 show that *backbone attacks* are highly effective at producing transferable adversarial samples regardless of the target model tuning *mode*, *dataset* or classification layer *depth*. This naive attack shows significantly higher transferability compared to a strong *black-box* attack with a sizeable query and iteration budget and almost all *proxy attacks*. The results are consistent across all configuration variations, showing that even a naive attack can exploit the target model vulnerabilities closely to a *white-box* setting, given the knowledge of the pre-trained backbone. Moreover, in Figure 4, we see that the adversarial samples produced from this attack, on average, cause a bigger shift in the model's decision-making

compared to *white-box attacks*. This indicates that backbone attacks amplify the uncertainty in the target model's predictions, making them more disruptive than conventional *white-box* attacks. A concerning aspect of backbone attacks is their effectiveness in resource-constrained environments. Unlike black-box attacks, which often require extensive computation or iterative querying, backbone attacks can be executed with minimal resources, leveraging pre-trained models freely available in public repositories. This ease of implementation raises concerns, as it lowers the barrier for malicious actors to exploit adversarial vulnerabilities.

## 5.3 KNOWING THE WEIGHTS VS KNOWING EVERYTHING ELSE

To isolate the impact of pre-trained backbone knowledge in adversarial transferability, we take two ResNet-50 SwAV backbones pre-trained with different batch sizes and further tuned with identical configuration variations. This allows for the production of two sets of models with matching training configurations but varying weights; one set is chosen as the target, and the other as the proxy model. We aim to compare the adversarial transferability of the attacks from the set of proxies towards their matching targets with the backbone attacks. This allows us to simulate conditions where adversaries either know all configurations but lack the weights or have access to the backbone weights alone.

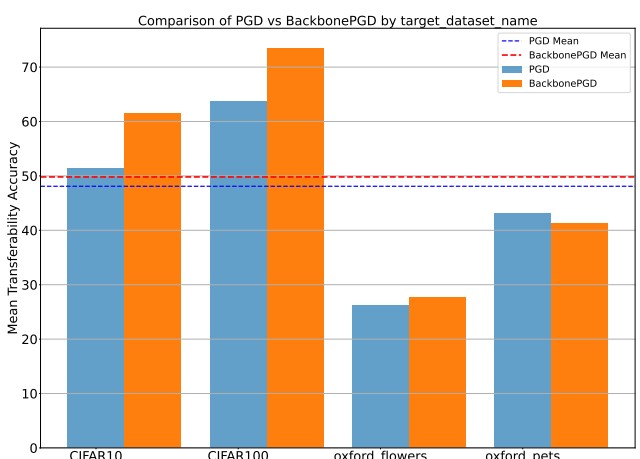

Figure 5: Scenarios where adversaries either lack backbone weights or only possess them. The latter is shown as *BackbonePGD* (SwaV ResNet-50).

Our results in Figure 5 show that the knowledge of the pre-trained backbone is, on average, a stronger or at least an equivalent signal for producing adversarially transferable attacks compared to possessing all of the training configurations without the knowledge of the weights. The results are consistent across all datasets, with domain-specific datasets showing marginal differences in adversarial transferability between the two scenarios. This means that possessing information about only the target model backbone is equivalent to knowing all of the training configurations for constructing transferable adversarial samples.

## 6 CONCLUSIONS

We investigated the vulnerabilities of machine vision models fine-tuned from publicly available pre-trained backbones under a formalised *grey-box* adversarial setting. We systematically measured the effect of varying levels of training configuration availability for constructing transferable adversarial attacks. We also explored a naive *backbone attack* method in this setting, showing that access to backbone weights is sufficient for obtaining adversarial attacks significantly better than query-based *black-box* settings and comparable to white-box performance. We found that these attacks often induce more drastic shifts in the model's decision-making compared to white-box attacks. We demonstrated that access to backbone weights is equivalent in effectiveness to possessing all tuning configurations about the target model, making public backbones a critical security concern. Our results highlight the risks associated with sharing pre-trained backbones, as they enable attackers to craft highly effective adversarial samples, even with minimal additional information. These findings underscore the need for more thought-out practices in sharing pre-trained backbones to mitigate the inherent vulnerabilities exposed by adversarial transferability.

ETHICS STATEMENT

We confirm that our experiments respect privacy, avoid misuse, disclose limitations and potential harms, and acknowledge societal impacts. All datasets used were obtained under proper licenses or permissions, and any use of adversarial methods is justified and documented to alert downstream users of risks.

REPRODUCIBILITY REPORT

To reproduce the results of our study, we provide the complete codebase, processing pipelines and hyperparameters for each dataset. We also make the rigorous details and checkpoints of all of the models in our study across all of the datasets publicly available for further experimentation and exploration.

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

| Model Families | CIFAR10 | CIFAR100 | Oxford Flowers | Oxford Pets |
|---|---|---|---|---|
| AlexNet (Colorization, IN1K) | 88.97 | 98.96 | 24.91 | 49.94 |
| AlexNet (Colorization, IN22K) | 89.56 | 98.92 | 25.19 | 50.06 |
| AlexNet (Colorization, YFCC100M) | 87.84 | 98.55 | 24.91 | 49.96 |
| AlexNet (Jigsaw, IN1K) | 53.25 | 74.03 | 26.96 | 45.38 |
| AlexNet (Jigsaw, IN22K) | 53.06 | 73.76 | 30.61 | 49.86 |
| AlexNet (DeepCluster V2) | 49.59 | 64.38 | 27.15 | 44.52 |
| ResNet-50 (Jigsaw, IN22K) | 61.03 | 81.81 | 26.37 | 47.28 |
| ResNet-50 (Colorization, IN1K) | 89.86 | 98.07 | 24.91 | 50.12 |
| ResNet-50 (Colorization, IN22K) | 88.99 | 97.89 | 27.01 | 50.00 |
| ResNet-50 (Jigsaw, IN1K) | 56.34 | 80.01 | 25.46 | 48.12 |
| ResNet-50 (Jigsaw, IN22K) | 54.48 | 75.08 | 26.79 | 47.75 |
| ResNet-50 (RotNet, IN1K) | 47.71 | 72.61 | 37.86 | 45.69 |
| ResNet-50 (Jigsaw, IN1K) | 58.02 | 78.32 | 26.17 | 48.06 |
| ResNet-50 (NPID) | 58.37 | 80.39 | 49.77 | 48.42 |
| ResNet-50 (PIRL) | 58.80 | 84.12 | 34.03 | 44.10 |
| ResNet-101 (SimCLR) | 55.09 | 70.34 | 28.54 | 47.12 |
| ResNet-50 (SimCLR) | 51.57 | 65.91 | 30.26 | 44.12 |
| ResNet-50 (SwAV, 400ep) | 48.63 | 68.46 | 28.79 | 44.33 |
| ResNet-50 (SwAV, 800ep) | 50.23 | 67.89 | 27.73 | 45.33 |
| DeiT-Small (DINO) | 63.37 | 85.08 | 26.56 | 47.26 |
| XCiT-Small (DINO) | 49.46 | 64.84 | 27.19 | 46.76 |

Table 2: Adversarial Transferability Averaged for each dataset per model architecture type

# A EXPERIMENTAL DETIALS

## A.1 MODEL VARIATIONS AND ADVERSARIAL TRANSFERABILITY

The adversarial transferability for each type of model can be seen in Table 2. The complete set of model variations used throughout the experimentations can be observed in Table 3.

## A.2 COMPUTATIONAL RESOURCES

All experiments were conducted using two compute nodes, each equipped with 8 NVIDIA A100 GPUs (80 GB memory per GPU), resulting in a total of 16 GPUs. Each node was powered 96 vCPUs (Intel Xeon Platinum) and 400 GB of RAM. Training all 352 model variations required approximately 3200 GPU-hours. The adversarial evaluation phase—including proxy attack generation, backbone attacks, and high-budget PGD experiments—required an additional 1800 GPU-hours. To ensure consistency, we fixed all random seeds to 42 across all runs, including for NumPy, PyTorch, and Python's built-in random module. Model tuning configurations, checkpoints, logs, and attack results were stored for full reproducibility.

| SSL Method | Pretraining Dataset | Architecture |
|---|---|---|
| **Colorization** (Zhang et al., 2016) | | |
| Colorization | YFCC100M | AlexNet |
| Colorization | ImageNet-1K | AlexNet |
| Colorization | ImageNet-1K | ResNet-50 |
| Colorization | ImageNet-21K | AlexNet |
| Colorization | ImageNet-21K | ResNet-50 |
| **Jigsaw Puzzle** (Noroozi & Favaro, 2016) | | |
| Jigsaw Puzzle | ImageNet-21K | ResNet-50 |
| Jigsaw Puzzle | ImageNet-1K | ResNet-50 |
| Jigsaw Puzzle | ImageNet-21K | ResNet-50 |
| Jigsaw Puzzle | ImageNet-21K | AlexNet |
| Jigsaw Puzzle | ImageNet-1K | AlexNet |
| Jigsaw Puzzle | ImageNet-1K | ResNet-50 |
| **PIRL (Jigsaw-based)** (Misra & van der Maaten, 2020) | | |
| PIRL | ImageNet-1K | ResNet-50 |
| **Rotation Prediction** (Gidaris et al., 2018) | | |
| RotNet | ImageNet-1K | ResNet-50 |
| **DINO** (Caron et al., 2021) | | |
| DINO | ImageNet-1K | DeiT-Small |
| DINO | ImageNet-1K | XCiT-Small |
| **SimCLR** (Chen et al., 2020a) | | |
| SimCLR | ImageNet-1K | ResNet-50 |
| SimCLR | ImageNet-1K | ResNet-101 |
| **SwAV** (Caron et al., 2020) | | |
| SwAV | ImageNet-1K | ResNet-50 |
| SwAV | ImageNet-1K | ResNet-50 |
| **DeepCluster V2** (Caron et al., 2018) | | |
| DeepCluster V2 | ImageNet-1K | AlexNet |
| **Instance Discrimination (NPID)** (Wu et al., 2018) | | |
| NPID | ImageNet-1K | ResNet-50 |

Table 3: Summary of Self-Supervised Learning Methods, Pretraining Datasets, and Architectures used in our study.

