# OpenReview forum: "With Great Backbones Comes Great Adversarial Transferability"
_ICLR.cc/2026/Conference — ICLR 2026 Conference Withdrawn Submission_

### Official Review · Reviewer_9B3r · 2025-10-26

**Soundness:** 1
**Presentation:** 2
**Contribution:** 2
**Rating:** 2
**Confidence:** 5

**Summary:**

This paper provide a detailed evalution on the impact of different levels of pre-knowledge of victim models, which are all regarded as grey-box scenarios in the paper. Results are provided competely in graphic presentation and indicate an intermediate performance between white box and black box attacks. The author also proposed a backbone attack which essentially use the identical pre-trained backbone and validated its effectiveness.

**Strengths:**

This paper provides a systemactic evaluation on the model hyperparameters in a finer perspective, including weights, trainind method, datasets, etc. These works validate some established consensus regarding adversarial attacks in an experimental way.

**Weaknesses:**

The paper suffers from several major weaknesses, which can be categorized into 3 aspects: **Contribution**, **Experiments**, and **Writing**.

1. **Contribution**. The research contribution of this work is severely limited. Specifically, the method proposed in this paper, i.e., backbone attack, has no methodological innovation as it simply uses PGD to minimize the cosine similarity between adversarial and original examples. Apart from the method, the innovation brought by this paper comes from the finer-grained hyperparameters regarding attacker knowledge (training, dataset, layer). In this regard, this paper more resembles a technical report in exploration for grey-box settings with the optimal transferability rather than a scientific paper. Furthermore, the paper cited an unpbulished paper ( Katzir & Elovici, 2021), which questioned the practicality of transfer attack. While such a statement remains questionable, the paper also failed to provide practical evaluations and demonstrate its supriority/contribution regarding this limitation.

2. **Experiments**. Experiments lack depth, consistency, and a significant number of details, making it difficult to convey the effectiveness of their proposed method and subsequently convince readers. *Depth-wise*, all experiments remain shallow as most of them, although presented as main results, are simple ablation studies for comparing different hyperparameters. Does the optimal design exhibit universality? Why do such settings prevail among others? Besides, only 4 classification datasets (CIFAR/Oxford) are considered in the paper, without using the predominant Imagenet family. *Consistency-wise*, the results repeatedly jump from ASR to TSR, entropy, and divergence, etc., without sticking to a main/crucial metric to demonstrate the transferability. For example, a source-target table that intuitively shows the white-box, grey-box, and black-box results simultaneously, using ASR, as is done in most papers on transferability. *Detail-wise*, many important details are missing or lack a specific explanation. For example, the number of steps, step-size, and epsilon for PGD, the model used as proxy/target model and why, the specific reason for choosing sqaure attack. These missing details make it difficult to understand the insight which authors try to convey.

3. **Writing.** The writing also further hinders reading. Despite some bizarre wording and repetitive notation, such as 'machine vision' (should be computer vision) and $ z(\cdot )$-$ f(\cdot) $, the paper lacks a tracable and self-contained storyline. To begin with, the motivation for adopting such a grey-box setting is unclear and unjustified given that transfer-based attacks have achieved promising results. The paper further proposes a frame work with several tunable hyperparameters for grey-box evaluation. Despite the scalability of such framework, the information presented is too scattered and lacks in-depth analysis/innovation. Specifically, some conclusions are even contradictory to the other: In line 375, the paper states that `the depth of the tuning is the least important knowledge for
obtaining a transferable attack`, while in line 419, the paper writes that `attackers do need to have access to ... model classification depth to craft adversarial samples`, which requires further clarification.

**Questions:**

See weaknesses.

---

### Official Review · Reviewer_zN99 · 2025-11-01

**Soundness:** 3
**Presentation:** 3
**Contribution:** 2
**Rating:** 4
**Confidence:** 4

**Summary:**

This paper investigates the adversarial robustness of models fine-tuned from publicly available pre-trained backbones (e.g., ResNet, ViT) under a “grey-box” setting, where attackers have partial knowledge of the target model (e.g., backbone weights, fine-tuning mode). The authors introduce a simple “backbone attack” method that leverages the shared feature extractor to generate transferable adversarial examples, and conduct an extensive empirical study (≈20,000 experiments) across multiple datasets and configurations. The paper concludes that access to pre-trained backbone weights is sufficient to achieve near white-box performance in transfer attacks.

While the paper is well written and the empirical analysis is extensive, the methodological novelty is limited, and several core findings overlap with conclusions already reported in prior works on transferable adversarial attacks. The work is primarily an observational study without introducing a truly new algorithmic insight or robust defense/attack strategy.

**Strengths:**

1. The topic is relevant and timely, especially given the prevalence of shared pre-trained models in modern vision pipelines.

2. The authors provide a large-scale, systematic empirical analysis across numerous backbone configurations, datasets, and tuning modes.

3. The experimental observations (e.g., backbone access ≈ full-knowledge access) are clearly presented and supported by data.

**Weaknesses:**

1. The proposed backbone attack is essentially a simplified variant of standard PGD that maximizes cosine distance in the feature space. While the systematic evaluation is valuable, the technical contribution is modest—no new attack or defense mechanism is introduced.
The paper’s main strength lies in empirical observations rather than algorithmic innovation. To improve impact, the authors could propose a new attack/defense method motivated by the findings (e.g., a method exploiting or mitigating backbone vulnerability), rather than stopping at descriptive analysis.

2. The central conclusion—that access to backbone weights suffices for strong transferability (“access to backbone ≈ white-box”)—has been noted in several earlier studies, including [1] [2]. These works have already reported that shared backbone or feature-space similarity drives high transfer success across tasks. The current paper does not sufficiently differentiate its contributions from these prior findings or discuss what new insights are obtained beyond broader benchmarking.

3.  The study focuses entirely on classification tasks using standard datasets (CIFAR-10/100, Oxford Pets, Flowers). This significantly limits the generality of the conclusions. Since many modern applications of pre-trained backbones involve detection, segmentation, VQA, and captioning, extending the evaluation to these domains would make the results more impactful and practically relevant.

4. While the large-scale empirical analysis is thorough, the paper does not propose actionable outcomes for the community, such as defense strategies or guidelines for safe backbone sharing. The study would be more valuable if the authors leveraged their observations to derive practical insights or quantitative mitigation recommendations.

[1] Transferable Adversarial Attacks on SAM and Its Downstream Models, In Neurips 2024.
[2] AdvCLIP: Downstream-Agnostic Adversarial Examples in Multimodal Contrastive Learning, In ACM MM 2023.

**Questions:**

Please see the weakness

---

### Official Review · Reviewer_PVk1 · 2025-11-01

**Soundness:** 2
**Presentation:** 3
**Contribution:** 3
**Rating:** 4
**Confidence:** 4

**Summary:**

The paper investigates gray-box adversarial transferability in realistic settings where attackers know only the publicly released backbone weights. It conducts a large-scale study across datasets, architectures, and fine-tuning regimes, and introduces a simple Backbone Attack that perturbs representations in the backbone feature space. Key findings show that adversarial examples crafted on proxy models outperform strong query-based black-box methods and approach white-box performance and the proposed Backbone Attack likewise exceeds black-box baselines and nearly matches white-box results.

**Strengths:**

1. Realistic gray-box formulation tied to today’s model-sharing ecosystem.
2. Large-scale, systematic study across many backbones/datasets with consistent metrics.
3. Backbone Attack is simple, reproduces easily, and approaches white-box.
4. Clear empirical insight: fine-tuning mode dominates transferability; backbone weights is equivalent in effectiveness to possessing all tuning configurations about the target model.

**Weaknesses:**

1. Black-box baselines: main-text details on query budgets/early-stopping are sparse; broader black-box comparisons would help, eg, transfer-based attack.
2. Limited to classification and small datasets; unclear if results hold for detection/segmentation or larger-scale datasets.
3. Reduced transfer on domain-specific datasets is noted but under-analyzed.

**Questions:**

1. What explains the performance drop on domain-specific datasets (class imbalance, background bias, scale)? Any mitigation (augmentation, cross-domain proxies)?
2. Does “backbone ≈ full config” still hold for deeper heads (>3 layers) or other SSL families?
3. Could you add practical guidance for safer backbone sharing (eg. minimum disclosure)?

---

### Official Review · Reviewer_1KqY · 2025-11-12

**Soundness:** 3
**Presentation:** 2
**Contribution:** 2
**Rating:** 2
**Confidence:** 3

**Summary:**

The paper investigates a grey-box threat model where attackers have partial knowledge of the target model’s construction (e.g., backbone weights, fine-tuning mode, dataset, depth) for the adversarial vulnerabilities of machine vision models fine-tuned from publicly available self-supervised learning (SSL) backbones. They have simulated over 20,000 adversarial transferability comparisons across 352 models from 21 backbone families, using various fine-tuning configurations and datasets. A key contribution is the “backbone attack”, which uses only the shared backbone (without tuning configuration knowledge) to generate adversarial examples via PGD in representation space. The authors show that this naive attack can outperform strong black-box attacks and approach white-box effectiveness. They also analyze which tuning configurations most affect transferability and model decision shifts.

**Strengths:**

1. It presents a backbone-only attack that is simple yet surprisingly effective, highlighting risks in model-sharing practices.

2. It covers a wide range of experiments, such as 352 models, 4 datasets (CIFAR-10/100, Oxford Pets, Flowers), multiple SSL methods (SimCLR, SwAV, DINO, PIRL, etc.), and attack types (PGD, FGSM, Square).

3. The paper is well-structured, with clear definitions of tuning configurations, proxy models, and attack metrics (ASR, TSR).

**Weaknesses:**

> The backbone attack is essentially PGD in representation space, similar to prior work in self-supervised adversarial training (e.g., NPR).

> The idea of transferability from shared components has been explored in surrogate-based attacks and meta-surrogates.

> While the paper exposes vulnerabilities, it does not suggest any concrete defenses or guidelines for secure backbone sharing.

> All experiments are on classification tasks. It’s unclear whether the findings generalize to other domains (e.g., segmentation, detection, multimodal models).

> The claim that backbone knowledge is “equivalent” to full configuration knowledge may be dataset-dependent. Domain-specific datasets (e.g., Oxford Flowers) show reduced transferability.

**Questions:**

1. How does the backbone attack compare to NPR and other representation-space attacks in terms of formulation and effectiveness?

2. Can you propose defense mechanisms or best practices for backbone sharing to mitigate the risks you expose?

3. Do your findings generalize to non-classification tasks (e.g., segmentation, detection)? Have you tested any?

4. Is there a lighter benchmark or subset of your framework that can be used for reproducibility by researchers with limited compute?

5. Can you clarify the failure modes of backbone attacks—when do they fail to transfer effectively, and why?

---

### Note · Authors · 2025-11-16

**Comment:**

We are withdrawing the submission because we are certain that at least two, and possibly three, of our reviews have been generated by AI. This is a very sad day for science and professional integrity.

**Withdrawal Confirmation:**

I have read and agree with the venue's withdrawal policy on behalf of myself and my co-authors.